# Self-reflection like Humans, Editable-LLM (E-LLM) is All You Need

**Yanchen Wu**
Student ID: 2023311818
wu-yc23@mails.tsinghua.edu.cn

**Gangxin Xu**
Student ID: 2023370010
xgx23@mails.tsinghua.edu.cn

**Dongchen Zou**
Student ID: 2023312975
zdc23@mails.tsinghua.edu.cn

## 1   Background

In recent years, Large Language Models (LLMs) have achieved significant advancements in the field of Natural Language Processing (NLP), largely driven by developments in deep learning architectures based on neural networks. Notably, the Transformer model has been at the forefront of these breakthroughs. From the early GPT series to BERT, T5, and more recent models like ChatGPT, LLMs have demonstrated remarkable performance across a wide range of NLP tasks, including but not limited to text generation, translation, question answering, and summarization. However, despite their impressive capabilities, generative LLMs still face inherent limitations.

One key shortcoming is that most LLMs generate the next token based solely on prior context, with no regard to the difficulty or complexity of the task at hand. The time taken for token generation is dependent on the length of the preceding text rather than the complexity of the reasoning required. Furthermore, LLMs often exhibit overconfidence in their outputs, attempting to maintain logical coherence by compounding errors with further incorrect information. This tendency can lead to outputs that are increasingly misaligned with the intended response, creating a "snowball effect" of misinformation.

While some researchers have attempted to address these issues by incorporating Chain-of-Thought (CoT) reasoning to extend the model's thought process, this approach still faces challenges. Designing effective CoT reasoning chains often requires substantial manual intervention, and the fixed structure of these chains may not be flexible enough to adapt to all tasks. Consequently, current LLMs struggle to effectively self-correct or engage in iterative reasoning when faced with complex problems.

Against this backdrop, we propose the Editable Large Language Model (E-LLM), which aims to address these limitations by allowing for more dynamic interaction with generated text. In E-LLM, the model is designed to not only append new tokens but also insert or delete tokens within the text, offering a level of flexibility that traditional LLMs lack. This capability enables E-LLM to perform on-the-fly error correction, avoiding the constraints imposed by prior outputs. Additionally, E-LLM can adjust its reasoning time based on the complexity of the task, offering more efficient and tailored responses. Unlike CoT approaches, E-LLM does not require manually designed reasoning chains, and it produces cleaner, more accurate outputs.

In summary, E-LLM offers several key advantages: it allows the model to self-correct during the generation process, reducing logical inconsistencies; it dynamically adjusts reasoning time based on task complexity, enhancing efficiency; and it eliminates the need for manually crafted reasoning chains, while still providing superior reasoning capabilities.

This novel approach to LLM design promises to address some of the fundamental bottlenecks faced by current generative models, offering greater accuracy, flexibility, and usability in NLP applications.

38th Conference on Neural Information Processing Systems (NeurIPS 2024).

## 2 Definition

Suppose that the output of the traditional LLM is an n-dimensional vector $X \in \mathbf{R}^n$. And we consider further processing of this output, that is defining an action matrix $Y \in \mathbf{R}^{m \times n}$ where $m$ represents an optional operation (such as adding or deleting), and $n$ represents the n-th element of $X$ being operated on. The elements in the matrix Y represent the signal strength of the operation, and the larger the value, the better the operation. Inspired by reinforcement learning, we will use methods such as Monte Carlo to estimate the matrix $Y$, and then guide our model to modify the output $\widehat{X}$.

## 3 Related Work

In recent years, Large Language Models (LLMs) based on the Transformer architecture have revolutionized Natural Language Processing (NLP), achieving remarkable success in tasks such as text generation, translation, and question answering. Notable models include OpenAI's GPT series [3] and Google's BERT [5], both of which have set new benchmarks in the field.

To enhance reasoning capabilities, researchers have increasingly adopted the Chain-of-Thought (CoT) prompting approach introduced by Wei et al.[12]. CoT encourages models to articulate their reasoning processes explicitly, allowing them to tackle complex tasks that require multiple steps of reasoning. While CoT has demonstrated effectiveness in improving LLM performance on challenging problems, it relies on manually crafted reasoning paths, which can limit its scalability and adaptability.

Despite these advancements, traditional LLMs exhibit inherent limitations, such as generating contextually coherent yet factually incorrect outputs. Holtzman et al. [9] highlighted how LLMs can propagate errors by maintaining overconfidence in previous responses. To address these challenges, various strategies have emerged, including reinforcement learning from human feedback [4] and self-distillation methods [7], which enhance output alignment and reliability. However, these approaches still lack the capability for real-time edits during text generation.

## 4 Proposed Method

For our proposed method, we tend to use widely recognized datasets. Based on prior studies and the availability of data, we plan to use the following datasets:

- LAMBADA [11]: This dataset measures the ability of language models to predict the last word of a sentence.
- MMLU [8]: We hope that the proposed method has a good generalisation ability, i.e., it performs well for different domains of the corpus.
- Chinese corpus: We also find the following open-source Chinese datasets: Baidu Knowledge knowledge-based datasets, Unicom Q&A data,financial industry Q&A data, insurance industry Q&A data, Agricultural bank Q&A data, and telecom Q&A data.

Next, we plan to compare our proposed method with baseline methods. These include state-of-the-art models that excel in language generation:

- GPT-4 [1]: As one of the most powerful models, GPT-4 achieves strong performance on benchmarks like MMLU, making it an excellent baseline.
- GLM-130B[6, 13]: This model surpasses GPT-3 on a variety of benchmarks, including LAMBADA, which is an appropriate baseline to assess the impact of draft-revision strategy.
- Claude 3 [2]: This model excels in nuanced content generation, reasoning, and coding tasks. It is a good baseline for evaluating improvements in reasoning quality.
- DeepSeek-V2 [10]: It performs well on language understanding tasks such as MMLU.

To implement our proposed method based on the dataset, we tend to follow the following steps.

Firstly, we will generate a draft based on the input sequence and use different models to continue editing based on that draft. Secondly, we will train our proposed model based on data from the selected datasets and optimize the token revision process. Eventually, we compare the revised drafts against the baselines mentioned using the selected datasets.

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

## A    Appendix / supplemental material

Optionally include supplemental material (complete proofs, additional experiments and plots) in appendix. All such materials **SHOULD be included in the main submission.**

