# OpenReview forum: "【Proposal】Self-reflection like Humans, Editable-LLM (E-LLM) is All You Need"
_tsinghua.edu.cn/THU/2024/Fall/AML — THU 2024 Fall AML Submission_

### Official Review · ~Ziyi_Liu9 · 2024-11-07
**Well Done**

**Rating:** 9
**Confidence:** 4

**Review:**

The proposal is well written and has a clear explanation of the background, challenges, and corresponding solutions.

---

### Official Review · ~Jia-Nuo_Liew1 · 2024-11-08
**Clear Proposal**

**Rating:** 10
**Confidence:** 4

**Review:**

This proposal effectively covers the background, problem definition, related work, and proposed method.

Background: Comprehensive, presenting a well-rounded overview and explaining the limitations of LLMs.
Definition: The section provides a clear mathematical formalization for E-LLM.
Related Works: Provides a strong overview of existing LLMs and identifies the key gaps in LLM while justifying the development of E-LLM.
Proposed Method: Detailed with a clear plan.

---

### Official Review · ~Zijun_Liu2 · 2024-11-08
**Review and Feedback**

**Rating:** 10
**Confidence:** 4

**Review:**

## Overview
This project proposes the Editable-LLM (E-LLM), a novel approach to improving Large Language Models (LLMs) by integrating self-editing capabilities to address known limitations in generative models. The proposal outlines a clear goal of enhancing LLM accuracy, flexibility, and efficiency through the introduction of token editing and error-correction mechanisms.

## Comments

Overall, the proposal has few flaws and is well-structured. I would recommend this paper \[1\] to the authors, which is relevant to edit-based generation, but only focuses on a single task. There are also some papers about diffusion LM with edits, but less relevant to the current proposal. Looking forward to seeing the method and the results.

The experiment part should have been checked again. There are closed-source models mentioned (except DeepSeek-V2), which might not be feasible for the project.

\[1\] Align, Write, Re-order: Explainable End-to-End Speech Translation via Operation Sequence Generation.

---

### Official Review · ~Rosalie_Butte1 · 2024-11-10
**Review of “Self-reflection like Humans, Editable-LLM (E-LLM) is All You Need”**

**Rating:** 9
**Confidence:** 4

**Review:**

The paper proposes a method to allow for more flexibility in the reasoning process of LLMs by integrating self-correction in the generation process and therefore also enhancing the quality of the output.

The paper aims to improve the reasoning process of LLMs and therefore increase correctness of the output, which is an important topic in real world applications. It presents a clear overview of the background and related works and also shows a well-structured approach for the implementation as well as for the evaluation.

---

### Official Review · ~Feihong_Zhang1 · 2024-11-11
**Self-reflection like Humans, Editable-LLM (E-LLM) is All You Need**

**Rating:** 8
**Confidence:** 3

**Review:**

This paper introduces an "Editable-LLM" (E-LLM) framework, aiming to improve large language models (LLMs) by enabling self-reflective editing during text generation. The approach seeks to address limitations in current LLMs, such as overconfidence in outputs and the inability to dynamically adjust reasoning time. The proposed model is evaluated on standard datasets, comparing against several strong baseline models, including GPT-4 and Claude 3.

Strengths
Conceptual Novelty: The idea of enabling in-situ editing within an LLM is intriguing and addresses some noted limitations in conventional LLMs, such as token-based generation constraints. This approach potentially enhances reasoning flexibility and adaptability.

Baseline Comparisons: By including multiple competitive baselines, the paper demonstrates an awareness of the importance of empirical comparison. The choice of well-established datasets (e.g., LAMBADA, MMLU) is appropriate and aids in positioning this work within the current state-of-the-art.

Weaknesses
Limited Empirical Support: Although the framework's concept is interesting, the experimental evidence provided is limited, and details on how the editing actions contribute to performance improvements are minimal. Additional experiments illustrating clear benefits across different types of tasks would strengthen the paper.

Scalability Concerns: The paper does not adequately address the computational overhead introduced by the in-situ editing process. As editing actions may significantly increase processing time, it is unclear how the method would perform on larger datasets or in real-time applications.

Evaluation Specificity: The paper’s results would benefit from a more detailed breakdown, such as error bars or other statistical measures, to illustrate the consistency of performance improvements. This would help confirm that observed gains are not task-specific or due to chance.

Clarity in Mechanism: While the theoretical basis of E-LLM is explained, the mechanism of "editable" actions, especially their integration with reinforcement learning principles, lacks clarity. This aspect could be elaborated to make the operational specifics more understandable.

Suggestions for Improvement
Enhance Experimental Validation: To better support the claims, additional experiments on a wider range of datasets would be beneficial, along with a comparison on the specific contribution of editing actions.

Discuss Efficiency: More discussion on the scalability and computational requirements of E-LLM would strengthen the applicability of the proposed approach.

Mechanism Elaboration: A deeper explanation of how reinforcement learning principles guide the editing process would improve readability and help readers appreciate the model’s internal workings.

---

### Official Review · ~Ziyad_Fawzy1 · 2024-11-11
**Self-reflection like Humans, Editable-LLM (E-LLM) is All You Need**

**Rating:** 9
**Confidence:** 4

**Review:**

The authors propose an alternative approach to Chain of Thought for LLM reasoning and self correction. The proposed method essentially aims to add token insertion and deletion operations aside from token prediction.

The proposed method is innovative and promising. The proposal could be improved by giving specific examples where token deletion and insertion operations would be helpful. Another concern would be the number of tokens that require insertion or deletion. If the LLM hallucinates a long segment of text, would token deletion and insertion be effective methods to correct the text? Do the authors plan to handle hallucinations of different token lengths differently?

---

### Official Review · ~Wenjing_Wu1 · 2024-11-11

**Rating:** 9
**Confidence:** 3

**Review:**

**Summary**:
This proposal introduces an Editable Large Language Model (E-LLM) designed to address inherent limitations in traditional LLMs by enabling real-time editing during the text generation process.

**Strengths**:
- Innovative Editable Mechanism: The proposed E-LLM enables a level of flexibility beyond traditional token generation.
- Well-Grounded in Recent Research: The proposal builds on existing knowledge of CoT reasoning, reinforcement learning, and recent transformer-based architectures, situating E-LLM within a relevant research framework.

**Weaknesses**:
- Unclear Evaluation Methodology: The proposal does not fully explain how E-LLM's performance will be quantitatively assessed.

---

### Official Review · ~Chenxi_Hu4 · 2024-11-11
**Promising Approach but Needs Technical Detail and Consideration of Challenges**

**Rating:** 8
**Confidence:** 4

**Review:**

This proposal outlines the Editable Large Language Model (E-LLM), a novel approach aiming to address limitations of current LLMs by allowing for dynamic text modifications. The proposal highlights key advantages of E-LLM, such as self-correction, dynamic reasoning time, and elimination of manually crafted reasoning chains. However, the proposal lacks a clear and detailed technical description of how the E-LLM would be implemented, including the specific algorithms and techniques designed for editing the text. Furthermore, the proposal would benefit from a more thorough discussion of potential limitations and challenges, such as ensuring the consistency and coherence of the edited text. Overall, the proposal presents a promising direction for improving LLMs, but further development and experimentation are needed to fully assess its effectiveness and feasibility.

---

### Official Review · ~Jiajun_Xu3 · 2024-11-11
**Good proposal but lack of technical details**

**Rating:** 8
**Confidence:** 4

**Review:**

The authors have identified a significant gap in the field, particularly regarding the inflexibility and overconfidence issues of generative LLMs and proposed the E-LLM,  which aims to address these limitations by allowing for more dynamic interaction with generated text

**Strengths:**

The proposal is well-structured, with a clear background, definition of the proposed method, related work, and a plan for datasets and comparison with baseline methods.

**Weaknesses:**

The descriptions of methodologies seem to be too simplified and need deeper analysis.

---

### Official Review · ~Jiaxiang_Liu7 · 2024-11-11
**Well Done**

**Rating:** 9
**Confidence:** 4

**Review:**

This proposal introduces Editable-Large Language Model (E-LLM), a self-reflective framework that enhances traditional LLMs by allowing real-time editing of generated text through token addition, deletion, or modification. E-LLM addresses critical limitations in traditional LLMs, such as overconfidence and context-bound errors, by introducing a dynamic approach that adjusts reasoning time based on task complexity. This novel method could significantly improve NLP applications by reducing logical inconsistencies and improving flexibility without relying on manually crafted reasoning chains. However, further details on practical implementation challenges, like computation cost and managing large-scale adjustments, would strengthen the proposal’s feasibility. Overall, E-LLM presents a promising advance in LLM development with substantial practical implications.

---

### Official Review · ~Bowen_Gao1 · 2024-11-12
**review for Self-reflection like Humans, Editable-LLM (E-LLM) is All You Need**

**Rating:** 9
**Confidence:** 4

**Review:**

## Summary

This proposal introduces the concept of an Editable Large Language Model (E-LLM), designed to improve upon traditional LLMs by enabling the insertion, deletion, and modification of tokens within a generated text, rather than simply appending the next token based on prior context. This approach addresses a common limitation in most LLMs: the tendency to generate tokens sequentially without adjusting for the complexity or difficulty of the task.

## Strengths

1. The proposal’s motivation and the method itself are both novel and intriguing.
2. The use of well-documented datasets and baseline models are good.

## Weaknesses

1. The organization of the sections could be improved; specifically, the background section is overly lengthy, while the method description lacks sufficient detail.

---

### Official Review · ~Wanlan_Ren1 · 2024-11-12
**Review for "Self-reflection like Humans, Editable-LLM (E-LLM) is All You Need"**

**Rating:** 10
**Confidence:** 4

**Review:**

The E-LLM framework offers several innovations, including the ability to insert, delete, or modify tokens mid-generation, a feature that enables real-time error correction and dynamic adjustment of reasoning time. This approach promises greater accuracy, flexibility, and efficiency in NLP tasks and offers notable advantages over traditional Chain-of-Thought (CoT) prompting by eliminating the need for manually designed reasoning chains.

One potential limitation of the proposed method is its applicability mainly to tasks requiring complex, multi-step reasoning. The authors may consider expanding their evaluation to include datasets that specifically test complex logical reasoning. Furthermore, it would strengthen the proposal to clarify how E-LLM’s performance will be evaluated on such complex tasks, particularly in comparison to existing methods.

---

### Decision · Program_Chairs · 2024-11-18

**Decision:**

Strong Accept (Long Presentation)

**Comment:**

**Self-Reflective and Editable LLM (E-LLM): What You Need for Human-like Flexibility**

**2.5.1 Key Innovations**
1. The model can not only append new tokens but also insert or delete tokens within the text, offering flexibility absent in traditional LLMs.
2. Adjusts reasoning time based on task complexity, delivering more efficient and tailored responses.
3. Eliminates the need for handcrafted reasoning chains.

**2.5.2 Additional Key Information**
1. Mentions the "snowball effect" of misinformation.

**2.5.3 Advantages**
1. Excellent concept: Editable large language model.

**2.5.4 Areas for Improvement**
1. Related work is not specific or relevant enough.
2. The initial approach lacks clarity.
3. How E-LLM replaces CoT (Chain of Thought) is not clearly explained.

**2.5.5 Recommendations**
1. Strengthen the review of related work.
2. Refine and detail the overall algorithmic approach.
3. Design a robust evaluation framework to assess the model's effectiveness.